# Emergency Departments as Care Providers for Patients with Cardiac Ambulatory Care Sensitive and Mental Health Conditions: Qualitative Interview and Focus Group Study with Patients and Physicians

**DOI:** 10.3390/ijerph19106098

**Published:** 2022-05-17

**Authors:** Martina Schmiedhofer, Anna Slagman, Stella Linea Kuhlmann, Andrea Figura, Sarah Oslislo, Anna Schneider, Liane Schenk, Matthias Rose, Martin Möckel

**Affiliations:** 1Departments of Emergency and Acute Medicine, Campus Mitte and Virchow-Klinikum, Charité-Universitätsmedizin Berlin, 10117 Berlin, Germany; anna.slagman@charite.de (A.S.); stella.kuhlmann@charite.de (S.L.K.); martin.moeckel@charite.de (M.M.); 2German Coalition for Patient Safety (Aktionsbündnis Patientensicherheit), 10179 Berlin, Germany; 3Department of Neurology, Charité-Universitätsmedizin Berlin, 10117 Berlin, Germany; 4Department of Psychosomatic Medicine, Charité-Universitätsmedizin Berlin, 10117 Berlin, Germany; andrea.figura@charite.de (A.F.); matthias.rose@charite.de (M.R.); 5Institute of General Practice, Charité-Universitätsmedizin Berlin, 10117 Berlin, Germany; soslislo@zi.de; 6Institute of Medical Sociology and Rehabilitation Science, Charité-Universitätsmedizin Berlin, 10117 Berlin, Germany; anna.schneider@charite.de (A.S.); liane.schenk@charite.de (L.S.)

**Keywords:** ambulatory care sensitive conditions, cardiac diseases, emergency department, frequent user, health care research, mental health conditions, qualitative research

## Abstract

Mental health conditions are frequent among patients with somatic illnesses, such as cardiac diseases. They often remain undiagnosed and are related to increased utilization of outpatient services, including emergency department care. The objective of this qualitative study was to investigate the significance of the emergency department in the patients’ course of treatment and from the physicians’ perspective. An improved understanding of the subjective needs of this specific patient group should provide hints for targeted treatment. This study is part of the prospective EMASPOT study, which determined the prevalence of mental health conditions in emergency department patients with cardiac ambulatory care sensitive conditions. The study on hand is the qualitative part, in which 20 semi-structured interviews with patients and a focus group with six ED physicians were conducted. Data material was analyzed using the qualitative content analysis technique, a research method for systematically identifying themes or patterns. For interpretation, we used the “typical case approach”. We identified five “typical patient cases” that differ in their cardiac and mental health burden of disease, frequency and significance of emergency department and outpatient care visits: (1) frequent emergency department users with cardiac diseases and mental health conditions, (2) frequent emergency department users without cardiac diseases but with mental health conditions, (3) needs-based emergency department users with cardiac diseases; (4) targeted emergency department users as an alternative to specialist care and (5) patients surprised by initial diagnose of cardiac disease in the emergency department. While patients often perceived the emergency department visit itself as a therapeutic benefit, emergency department physicians emphasized that frequent examinations of somatic complaints can worsen mental health conditions. To improve care, they proposed close cooperation with the patients’ primary care providers, access to patients’ medical data and early identification of mental health conditions after cardiac diagnoses, e.g., by an examination tool.

## 1. Introduction

Mental health conditions (MHCs), which refer to abnormal states of mental health [1], such as symptoms of depression and/or anxiety are widespread, both in the general population and among patients with somatic illnesses such as cardiac diseases (CDs) [2,3,4,5,6]. MHCs have been shown to impair health-related outcomes such as morbidity, mortality and quality of life [7,8,9,10,11]. A particular problem is that MHCs often remain undiagnosed in patients with somatic symptoms [12,13]. Moreover, MHCs have been associated with increased direct and indirect costs in the health care system [14,15,16] and increased utilization of outpatient services, including emergency departments (EDs) [5,17,18]. As a result, ED crowding has become a major public health issue worldwide, as it has been linked to deteriorating medical care and patient health outcomes. Of major concern are ambulatory care sensitive conditions (ACSCs), which comprise a number of diagnoses which often emerge in ED care and might also lead to hospitalizations, but are at the same time defined as conditions in which an ED or inpatient treatment could have been avoided by adequate and timely treatment in the outpatient sector, given appropriate and continuous care [19,20,21,22]. Furthermore, ED care focuses on the examination and treatment of acute complaints and often separates physical and mental health problems. This may lead to recurrent ED and ambulatory care presentations, when mental health problems remain undetected. Patients with MHCs often develop a need for urgent medical attention. They receive prompt treatment in the ED because patients with cardiac complaints are admitted to an urgent triage category [23]. Previous research has shown a relation between MHCs and CDs [24]. Acute anxiety conditions could also be experienced in patients with acute coronary syndromes such as palpitations and chest pain [25], while depression is connected to poor medical adherence in patients with CDs [26]. These comorbidities pose diagnostic challenges to ED examinations and sustainable treatment. Previous studies focused mainly on younger patients with MHCs and cardiac complaints [27,28,29,30]. However, more information is needed on older ED patients with cardiac complaints as the probability of CD increases with age [31]. In consequence, MHC comorbidities may remain undiagnosed. To assess the proportion of ED patients who suffer from acute cardiac symptoms and symptoms of MHCs, the quantitative part of EMASPOT examined the associations between MHCs and increased ED use as well as between MHCs and the occurrence of ACSCs. The screening of a representative ED patient cohort with cardiac complaints identified a total of 28.4% with current MHCs [32]. In light of these findings, complementary qualitative data could provide deeper insight into patients’ perspectives and motivations for visiting the ED and thus elucidate the role of the ED within their course of treatment [14,33,34]. Moreover, the patients’ view could provide essential information for the development of patient-centered, improved health care services [33]. In addition, patient demands should be explored from a professional perspective of ED physicians to develop ideas for improved, targeted care.

### Objective

The aim of the study on hand is to gain a deeper understanding of the subjective context in which patients with cardiac ACSCs seek help in the ED and to analyze possible differences in their approaches. In addition, patient demands should be explored from a professional perspective of ED physicians to develop ideas for improved, targeted care. To achieve the study objective, we defined the following research questions:What different types of patients can be derived with regard to their subjective motivation, personal background and perception of the role of the ED?Which measures would improve the delivery of care from ED physicians’ perspective based on their everyday experiences with patients presenting with cardiac ACSCs and assumed MHCs?

## 2. Materials and Methods

### 2.1. Design

The current study was part of the research project EMASPOT, a prospective multicenter cohort study as part of EMANet [35]. Patients were consecutively recruited during treatment in one of the eight EDs in the city center of Berlin, Germany, from June 2017 to September 2018. Inclusion criteria were (1) age ≥ 50 years; (2) multimorbidity ≥ 2 chronic diseases; and (3) cardiac complaints indicating ACSCs, such as angina, heart failure, atrial fibrillation and flutter, acute coronary syndrome and hypertension. Exclusion criteria were cognitive impairment; legal guardianship; or insufficient language skills in German, Turkish, Russian or English.

EMASPOT included both quantitative and qualitative data. Socio-demographic information and patients’ mental health status were assessed in an interview during the ED stay. Medical characteristics were extracted from the patients’ medical records as part of the quantitative survey. MHCs were measured with the well-established Patient Health Questionnaire (PHQ) for depression, anxiety, panic disorder and generalized anxiety disorder (GAD) [36,37,38]. The main objectives of EMASPOT were (1) to assess the prevalence of comorbid psychosomatic conditions in patients presenting to the ED with cardiac ACSCs, (2) to examine the influence of these conditions on the course of these patients during follow-up and (3) to evaluate expectations and barriers of patients towards ED care in order to derive approaches for optimized treatment models [13].

The study on hand used an embedded mixed methods approach [39,40] and carried out in-depth qualitative interviews with a subgroup of the quantitative cohort to make the quantitative results more comprehensible. A focus group interview with ED physicians was subsequently conducted [41].

Ethical approval was granted by the ethics committee (Charité (EA1/363/16)).

### 2.2. Data Collection: Patient Sampling

Between December 2017 and July 2018, one author (MS) who is experienced in qualitative research in health care settings carried out a purposive sample with *n* = 20 semi-structured face-to-face interviews with patients (interview guide translated into English in Table A1 Appendix A). All interviewees had participated in the quantitative EMASPOT survey during a preceding stay in one of the participating eight EDs. A study nurse, who had already explained the study goal in detail, asked about the willingness to take part in a subsequent qualitative interview. The majority of patients agreed. Of these potential participants, 27 were invited by phone to a personal interview. In order to present a broad variation of data, the interviewer approached potential participants with regard to heterogeneity in index clinics, age, gender and main complaints (Table 1) and explained the study goal. Of the requested possible participants, seven declined, mostly due to time constraints or because they did not feel well. The place and time of the interview were set by the respondents. According to personal preference, *n* = 11 interviews were carried out at patients’ private homes (*n* = 9) or workplaces (*n* = 2), at the interviewers’ office (*n* = 9) or during a subsequent ED visit (*n* = 1). One interview with a participant who lived at a far distance was conducted by phone. Before the data collection began, all participants provided written informed consent. The interviewer emphasized that participation was voluntary and could be withdrawn at any time. Interview duration and thematic depth were determined by the respondents. The interviews took from 15 to 52 min with a median duration of 30 min. Following each interview, field notes were taken to document impressions of atmosphere, nonverbal communication and special features for confirmability. All interview and field notes were transcribed verbatim and entered into the qualitative data software MAXQDA2020 anonymized.

### 2.3. Data Collection: Physician Sampling

After the first evaluation of patients’ interviews, a focus group with a stratified sample of six ED physicians from four study sites was carried out by two researchers with a background in public health and sociology (MS, SO) and one study assistant (interview guide translated into English in Table A2 Appendix A) in March 2019. The participants were approached by email or phone with regard to heterogeneity in index clinics, gender, occupational experience and professional position (Table 2). The focus group was conducted as an expert interview and took place in a conference room in one clinic and lasted one hour [42]. Before starting the discussion, all participants gave written informed consent.

### 2.4. Data Analysis

A qualitative content analysis (QCA) approach was chosen to reach the study goal and to answer the research questions [43]. QCA is a suitable and transparent method for descriptive qualitative data (see Table 1 for an example of data). The results were worked out by assigning verbatim units of the interviews both inductively to emergent and deductively to predefined categories. In the first stage, one of the authors (MS) reviewed the transcripts and coded them line by line. Then, after several discussions within the research group with professional backgrounds in sociology, psychology and public health (MS, AF, SLK, SO), a coding framework was created. Finally, the main dimensions of the coding structure were condensed to relevant aspects. A “typical case” approach was used to analyze the patient interviews [44,45]. Derived from the coding structure, the main narratives about the importance of the ED for health care were identified by condensing meaningful text units into common and distinct patterns derived from the interview data. On this data basis, a matrix was created with the respective variation of cardiac diseases and MHCs and the frequency, appreciation and perceived quality of ED and outpatient care (Table 3). By analyzing the content relations and grouping similarities and differences into common and distinct patterns, five different “typical cases” were built. By means of the QCA, the data of the physician focus group were analyzed concerning the improved treatment of patients with MHCs. All cases and further results are described in detail and highlighted with meaningful quotes.

#### Participants

An overview of the demographic data of the participants is presented in the Table 1 and Table 2.

## 3. Results

The aim of the study on hand is to gain a deeper understanding of the context in which patients with cardiac ACSCs seek help in the ED and to analyze differences in their approaches. In addition, patient demands should be considered from a professional perspective in order to develop ideas for better and targeted care. To achieve the study objective, our research questions asked about the different types of patients regarding their subjective motivation and perception of the role of the ED. Furthermore, we explored possible measures from physicians’ experiences which would improve the delivery of care for patients presenting with cardiac ACSCs and assumed MHCs.

Patient types have very different needs against the background of their course of disease(s) and treatment experiences (Section 3.1.1). Furthermore, patients’ data highlight the impact of comorbidities and MHCs on the perception of the ED as a site of rescue (Section 3.1.2) which at the same time is intertwined with reported satisfaction with outpatient care and management of the CDs (Section 3.1.3). In addition, we present missing organizational support and suggestions for outpatient care improvement from the patient respondents’ view (Section 3.1.4).

The results of the focus group with ED physicians mainly refer to patients with assumed MHCs. All discussants were familiar with this type of patient and reported dissatisfaction on the side of patients and professionals because the exclusion of an acute episode does not address an underlying MHC (Section 3.1.6). However, the ED physicians stated the lack of resources to examine an assumed MHC (Section 3.1.7) and brought up several suggestions to improve the delivery of care (Section 3.1.8), e.g., an examination tool to assess MHCs and refer patients to efficient treatment (Section 3.1.9).

In the following, the results are reported in detail.

### 3.1. Interviews with Patients

#### 3.1.1. Patient Types Regarding Motivation and Perception of ED Usage

To answer the first research question, a “typical case approach” was used to differentiate between patients with regard to their subjective motivation and perception of the role of the ED in their course of treatment. The typification is based on the data from the patients’ survey, interview narratives and medical record data. To structure the findings, a matrix was built to distinguish the cases from each other in their approach to the ED, the usage of outpatient care and/or their medical characteristics. The matrix can be found in Table 4, and the case descriptions, including meaningful quotes, can be found in Table 5.

In addition to characterizing patients’ motivations for seeking ED care, we also found patterns of behavior and perceptions that highlight patients’ needs and conditions, which are presented below.

#### 3.1.2. Impact of Patients’ MHCs on the ED as a Site of Rescue

Concerning the impact of MHCs on ED use, differences between patients with and without MHCs were identified. As shown above, patients with anxiety symptoms (Type 1 and 2) appreciate the ED as a safe place of rescue. When they are overwhelmed by emerging cardiac complaints, they head to the ED. According to their narrative, even the arrival at the ED as a site of safe care serves as a therapeutic purpose. Since they use the ED more than five times per year, they are designated as frequent users (FUs) [22]. Some patients diagnosed with heart disease reported how cardiac complaints alternate with other symptoms such as recurrent back pain or complications after a cancer surgery, as the following quotes illustrate:
“The heart attack came first and then the back. (…) And every now and then the back comes back, because of the heart I pushed it back. But now that the heart is all right again, I do notice the back again from time to time” (P12, male, 67 years).
“Then I was lying there after a serious operation [prostate carcinoma] and about two days later I had a very serious attack of heart arrhythmia. It was all very exciting, so I even came from the urology department to the cardiology department, which is also part of my medical history, always somehow surrounded by cancer and heart problems, so it’s not very nice, but this is how it looks for me, yes” (P03, male, 66 years).

In contrast, patient types with a pragmatic or targeted (Type 3 and 4) approach to the ED appreciated the medical service, but reported the discomfort of long examination times, lack of privacy or a noisy environment as reasons to avoid ED use as long as possible, as one quote highlights:
“There isn’t always the possibility to lie somewhere quiet (…) You lie there and have this bumpy heart rhythm yourself and then it beeps all the time” (P01, female, 60 years).

#### 3.1.3. Perceived Importance of the ED within Their Overall Course of Outpatient Care

Data show that ED care is more important when satisfaction with outpatient care is low and the CD is not well managed. While participants with MHCs highly appreciated the ED as a site of safe care, others used it in the case of exacerbation, but not in the course of regular care. FUs saw several PCPs on a regular or irregular scheme and highlighted the ED as a complementary provider:
“Well, over the years I’ve always had a double pack of doctors, I don’t just have one orthopedist, I have two orthopedists. I had the third slipped disc last year (…) So the pain center and the neurologist, I have practically everything twice, except for the GP (…) but yes, if the other parts don’t work, then the ED is certainly the place to go” (P08, female, 52 years).

#### 3.1.4. Missing Support and Improvement Suggestions from Patients’ Perspectives

Complaints were about barriers to making timely appointments with PCPs and a perceived lack of empathy. Furthermore, a subjective distrust of PCPs’ competence was reported. Causes of the complaints can lie both in objective problems with the availability of PCPs and in patients’ subjective feelings of insecurity. To receive specialist care after a cardiac diagnosis, participants stressed the challenge of making necessary short-term appointments. In some cases, a lack of information sharing between all PCPs was reported, although it remains unclear to what extent patients are informed about the actual contact between their care providers, as two quotes highlight:
“So at the GP I am told, ‘I can’t do anything for you’ and then I’m shown to the door. He says I’ve got three minutes for you” (P08, female, 52 years).
“Sometimes I have the suspicion that doctors don’t listen or don’t take you seriously. When you sit with a neurosurgeon, you tell him everything and he asks you three times, ‘What can I do for you?’” (P15, female, 58 years).

#### 3.1.5. Focus Group with ED Physicians

To answer the second research question, a focus group with ED physicians was conducted after the patient interviews were finished. Patients with MHCs were put in the thematic center of the focus group, as this topic was of specific interest (Appendix A Table A2). The results are represented by original quotes from the focus group participants to justify the interpretation of the data.

#### 3.1.6. Professional Experiences with Patients Presenting with Cardiac ACSCs and Assumed MHCs

All discussants were familiar with patients who present with cardiac complaints and leave after exclusion of an acute incident. According to the triage system, they are seen with high priority by ED physicians, as patients also mentioned positively. However, interviewees described that in patients’ view, the exclusion of an acute incident, a “non-diagnosis” (Phys_F) is often not taken as good news, because: “The patient is really happy when he goes home with a diagnosis” (Phys_C), which means: “You actually send them away with the awareness that they will either go to another ED (…) or attend someone else with the same complaints and the story often starts all over again” (Phys_E).

#### 3.1.7. Perception of Medical Treatment for This Group of Patients

The participants contributed several examples where both professionals and patients are left with the feeling of insufficient treatment. ED physicians indicated that they do not have enough time for addressing MHCs in patients, even though frequent examinations may worsen MHCs. They sometimes sense the patients’ underlying need to manage a comprehensive treatment: “My feeling is often, that they have a strong need to have [the complaints] managed by one person. But we certainly can’t do that in the ED and this is disappointing, yes” (Phys_D).

As described in the patients’ results section, the ED visit itself may achieve a short-term therapeutic benefit after a positive rescue experience: “It often happens that patients come with severe chest pain, shortness of breath, palpitations, etc., and as soon as they are here, they say: ‘now I am actually symptom-free’” (Phys_C). Such experiences were described as possibly leading to extremely frequent ED use, highlighted by the example of one well-known patient: “there are weeks when she really calls the paramedics ten times a day” (Phys_Q). Frequent ED use was seen as a potential contributor to an unfavorable disease course: “If you already know that there is somehow a psychosomatic component, then basically this psychosomatic disease has become even worse, right? And somehow it wasn’t really helpful for the patient that he had so many doctor contacts again. Yes, but nevertheless our task in the ED is to take chest pain seriously” (Phys_B).

#### 3.1.8. Necessary Measures and Resources to Improve the Delivery of Care

Several suggestions, addressing both ED treatment and PCP care, were made. To prevent the development of MHCs after myocardial infarction, targeted health training at an early stage, e.g., during rehabilitation, was proposed: “You observe many heart attack patients who present again shortly, (…) where one gets the impression that the time of the follow-up treatment (…) was not enough to deal with the topic and they are anxious, what could actually be prevented in an early phase” (Phys_B).

A further issue was a closer collaboration with patients’ PCPs. Positive experiences were reported and seen as an option for a comprehensive treatment: “So, when you have started this, GPs themselves call you more often, to give their own assessment in advance” (Phys_D). However, personal exchange with PCPs is described as limited to the opening times of practices and time constraints. To bypass the timely exchange in person, data access to patients’ medical history and recent treatment was also seen as a possibility for improving care.

#### 3.1.9. Examination Tool to Assess MHCs

The introduction of an examination tool to assess MHCs was brought in by the first author. The idea was perceived as promising for a sustainable and efficient treatment process: “…to make a first impression yourself, even as a non-psychiatrist or non-psychosomatist (…) that’s actually quite good, since my experience is, it’s better to go into more detail. If they hear ‘*a loony bin*’ at worst, that is actually wrong” (Phys_D).

In addition, the investment is seen as very beneficial for both patients and providers: ”If you take a little time to ask: ‘Is there a specific problem, that depresses you or that stresses you out?’ (…) I think you save a lot of time and resources if you take this time whenever it’s possible and perhaps also work out together with the patient whether there is a possible other cause” (Phys_D). However, time resources are needed to make a tool work: “You can’t just briefly ask that with two introductory sentences” (Phys_E) and: “when one ambulance comes in after the other, there isn’t enough time (…) because the acutely life-threatening patients are simply sicker” (Phys_B).

Participants summarized that an assessment tool only works effectively if subsequent treatment options can be offered. Two opinions emerged about the level of guidance patients need to accept MHC follow-up treatment: Some participants would be willing to schedule appointments through ED staff to avoid losing patients: ”in the nirvana of missed specialist appointments” (Phys_B), while others stressed patients’ self-responsibility in order to sustain their motivation. In the end, it was agreed that subsequent MHC treatment offers have to suit different personal needs.

With regard to the clinical implications of the frequency of cardiac patients with comorbid MHCs, EMASPOT II is currently developing and testing a psycho-cardiological training program for ED physicians and nurses to raise awareness of mental health issues related to cardiac symptoms [46]. Patient-oriented interventions could include educational information about the bidirectional relationship between physical and mental health problems. It seems promising that both ED staff and patients may benefit from such interventions in the ED setting.

## 4. Discussion

The aim of the study on hand was to gain a deeper understanding of the context in which patients with cardiac ACSCs seek help in the ED and to analyze possible differences in their approaches. To triangulate patients’ results with a professional point of view, a subsequent focus group interview with ED physicians was carried out, in which suggestions for optimized treatment were discussed. The discussion starts with the demands of “typical patients” which EDs are faced with. As presented in Section 3, the significance of ED treatment within their courses of disease differs. Patient data are mirrored and compared with the results from the ED physicians’ focus group interview. Subsequently, patients’ needs as well as suggestions of ED physicians for improved care are classified into the requirements of more patient-oriented ED structures.

### 4.1. Typical Cardiac ACSC Patients

Patients with chest pain are assigned to a high treatment urgency in the ED. However, as displayed in our data, they comprise various types of patients with regard to the severity of disease, medical urgency and motivation for and perception of ED usage. In the following, findings are discussed from the perspective of patients’ expectations for ED treatment.

### 4.2. Patients with (Co)morbid MHCs (Types 1 and 2)

Patients with MHCs (with or without diagnosed CDs) frequently suffered from fear of a serious heart incident. They attended several PCPs regularly. Regardless of the perceived quality of PCP care, they valued the ED as their point of rescue in case of upcoming emergencies. For them, the visit to the ED was successful because the mere arrival provides the certainty of being in a “safe” place that reduces pain and anxiety. Focus group participants confirmed these frequent phenomena of spontaneous relief, but they assessed the exclusion of an acute incidence as only partly successful when the MHCs could not be addressed. They described the EDs’ focus on caring for somatic complaints as a professional dilemma, because frequent somatic examinations may trap patients into an ED and PCP presentation circle, which can even worsen the MHCs [47]. Therefore, patients with and without cardiac diagnoses must be considered differently. Those who have suffered a serious heart disease, e.g., cardiac infarction, are at higher risk of developing MHCs after the incident, as several studies have shown [9,18]. ED physicians encountered this type of patient frequently and saw the cause in insufficient prevention of MHCs during the rehabilitation after the cardiac event.

Patients without heart disease (Type 2) seemed to integrate the frequent utilization of EDs into their course of life and “benefited” from the 24/7 accessibility of EDs while searching for sustainable help. As frequent somatic examinations contribute to short-term reliefs, they described ED visits as a coping strategy. However, when underlying MHCs cannot be addressed, the causes of the somatic complaints remain unattended [47].

### 4.3. Patients with Cardiac Diseases and Pragmatic (Type 3), Targeted (Type 4) or Unexpected Approach (Type 5) to ED Utilization

These patient types presented with acute cardiac ACSC complaints or exacerbations to the ED. By definition, ACSCs are acute or chronic disorders that could have been controlled or prevented by PCPs [12]. Therefore, the occurrence of ACSCs can be seen as an indicator of the quality of the overall health care system [48]. For that reason, the provision of outpatient treatment could critically be evaluated, taking into account that some of the acute conditions might not have been preventable in patients. Some patients were referred to the ED by their PCP, while others could no longer wait for an outpatient appointment or used the ED with a targeted approach as their exclusive cardiac care facility. The majority of interviewees were used to the ED setting in the course of their medical treatment. All of them valued the high medical standard and in turn accepted an uncomfortable setting, whereby the reported quality varied and depended on the degree of perceived ED crowding. The time period in which patients were diagnosed with cardiac disease ranged from the first time in the study ED to decades earlier.

Against this background, study data reflect the wide range of requirements EDs face from patients with cardiac ACSCs: ED staff has to communicate the initial diagnosis, carry out routine examinations and organize or coordinate further treatments. Some patients described the interface with outpatient care as challenging, especially the ones who left the ED with an initial cardiac diagnosis or those who were waiting for the next routine appointment. Based on nationwide population and hospital data, a German study analyzed factors influencing the development of ACSCs in patients with congestive heart failure, angina pectoris and arterial hypertension. While the density of PCPs was associated with a small (0.1–0.5%) reduction in cardiac ACSCs, the highest positive correlation to CDs was found with higher age, the group in which our study participants belong (0.7–3.6%) [49]. The patient–PCP ratio in Germany is higher than that in most other countries [50]. However, the German health care system, where inpatient and outpatient treatment are strictly separated and patients are free to choose their PCP and specialist, may contribute to ACSC development when patients choose the timely and best-equipped care at EDs instead of continuous outpatient care, where MHCs could be addressed in a regular treatment scheme. In turn, PCPs may refer time-sensitive and medically challenging patients to the ED [51,52]. However, the ED visit is only one facet of the chronic disease trajectory of patients. Singular contacts with ED physicians and the lack of close collaboration with PCP care may deteriorate the course of disease.

### 4.4. Improvement of EDs to Patient-Oriented Structures

As presented in Section 3, ED staff is faced with cardiac ACSC patient demands which they can only meet inadequately or in the short term. In the focus group, ED physicians revealed dissatisfaction with the insufficient treatment of patients with MHCs whose focus on somatic complaints was attributed to worsening the disease. These findings are in line with the results of previous research, stating that after a diagnosis of non-cardiac chest pain, distress may persist [47]. Most likely, patients are not referred to a mental health specialist and therefore symptoms such as anxiety cannot be adequately addressed and alternative, long-term coping strategies cannot be learned. To provide a more in-depth examination of MHC (co)morbidities, a tool to detect MHCs in the ED was advocated. However, participants underlined that the deployment of such a tool would probably require more than two or three brief questions. Instead, a thorough approach would be needed to achieve patients’ willingness for further treatment.

Two requirements were proposed to provide sustained treatment: close collaboration between PCP and ED physicians and access to current data on the patients’ treatment plans. Such information would facilitate anamnesis in the ED and save time for a thorough physician–patient conversation. In addition, reference to ongoing treatment and preliminary ED or PCP presentations could prevent patients from bypassing targeted treatment by presenting at different health care sites with the same complaints. Given the local density of EDs and PCPs in Berlin, such a strategy would be easier to achieve in urban than in rural regions. Therefore, a closer cooperation between all EDs in Berlin-Mitte could improve patient-centered care as a first step.

### 4.5. Clinical Implications

The causes of increased ED use and consequences such as crowding have been widely discussed for decades [18]. Unlike in countries with insurance-related barriers, the costs of health care are covered for all German inhabitants by mandatory sickness funds. The individual choice of health care providers is free of charge. Furthermore, the German health care system is well equipped with hospitals, specialists and PCPs [50]. Therefore, the reasons for ED admissions have to be considered in the outpatient care delivery structure. The increased presentation of older patients with complex and chronic conditions and ACSCs is one driver of ED crowding, as they require a comprehensive and time-consuming examination. Recent research found how older patients lack comprehensible information about their health state after discharge and are recommended to ED care in spite of outpatient options [53].

The share of ED patients presenting with cardiac complaints is about 11.5% [54]. The major finding of the EMASPOT study is that a substantial proportion of 28.4% of the patients with cardiac symptoms suffered from a comorbid MHC at the time of the ED visit, comprising moderate to severe symptoms of depression (23.3%), generalized anxiety disorder (12.2%) and panic disorder (4.7%) [32]. In order to address their needs with targeted solutions, this study was undertaken.

In recent years, the need for improvement of ED structures reached the German health policy, and several proposals have been discussed. One prominent concept is to reduce the number of ED patients through steered access after a brief assessment of medical urgency by a GP [55,56,57]. Without discussing the general feasibility here, it is very unlikely that patients with acute chest pain who are often admitted by paramedics will be excluded from immediate examination in the ED. Considering the substantial number of patients with cardiac ACSCs and the narratives of ED physicians, EDs should be provided with resources for sufficient treatment to approach patients with MHCs successfully and sustainably. Therefore, an infrastructure should be built to offer subsequent treatment options and reliable references to psychological or psychiatric consultation. Furthermore, a committed cooperation between EDs and PCPs is necessary for patient-oriented care. In order to achieve this, the multi-faceted tasks of EDs, e.g., substituting outpatient care, are to be recognized as part of acute care provision.

### 4.6. Limitations

Qualitative analysis is subjective by nature. Although measures were undertaken to reduce interviewer bias, they cannot be completely excluded. We did not manage to interview patients over 77 years of age and thus may not have included the perspectives of an older population. Furthermore, the study took place in an urban region with a high density of EDs and PCPs. The results cannot claim generalizability for rural areas. Therefore, further research is needed to obtain results from rural areas and to evaluate the extent to which the application of screening tools for MHCs improves patient care.

## 5. Conclusions

Our study shows that very different types of patients present to the ED with cardiac complaints, and many of these patients suffer from (co)morbid MHCs. Patients with cardiac complaints are treated with high urgency and are examined immediately. That is, for patients with diagnosed and undiagnosed MHCs, frequent treatment in the ED can exacerbate MHCs when they enter a vicious cycle of complaints–exclusions–complaints. As one consequence, ED personnel should be provided with a tool to identify MHCs in patients and explain the bidirectional relationship between physical and mental health problems. Furthermore, improved organizational health literacy of outpatient and inpatient providers could contribute to better patient outcomes and therefore reduce ED care.

## Figures and Tables

**Table 1 ijerph-19-06098-t001:** Characteristics of participating patients.

	Male *n* = 10	Female *n* = 10	All *n* = 20
**Age** in years (median) Min–Max	67 (50–74)	60 (51–77)	63 (50–77)
**Migrant (first generation) (*n* (%))**	1 (10)	1 (10)	2 (10)
**Educational Degree**			
Academic Degree (*n* (%))	4 (40)	2 (20)	6 (30)
Vocational Degree (*n* (%))	5 (50)	7 (70)	12 (60)
No Occupational Degree (*n* (%))	1 (10)	1 (10)	2 (20)
**Occupational Status**			
(Disability) Pensioner (*n* (%))	7 (70)	6 (60)	13 (65)
Employed/self-employed (*n* (%))	2 (20)	3 (30)	5 (25)
Job-seeking/unemployed (*n* (%))	1 (10)	1 (10)	2 (10)
**Personal Status**			
Living with spouse (*n* (%))	7 (70)	7 (70)	14 (70)
Single (*n* (%))	3 (30)	3 (30)	6 (30)
**Medical Characteristics *** 8 participants had one and 12 participants between 2 and 5 diagnoses			
Coronary heart disease (*n* (%))	3 (30)	3 (30)	6 (30)
Heart failure (*n* (%))	1 (10)	4 (40)	5 (25)
Cardiac arrhythmia (*n* (%))	9 (90)	5 (50)	14 (70)
Arterial hypertension (*n* (%))	7 (70)	10 (100)	17 (85)
Subacute myocardial infarction (*n* (%))	-	1 (10)	1 (5)
**Mental Health Condition** (*n* (%)) (PHQ9 > 9 and/or GAD7 > 9 and/or PHQ_PD and/or Panic and/or PHQ_Anxiety) *	3 (30)	6 (60)	9 (45)
**Psychotherapeutic Treatment ***			
Yes, ever (*n* (%))	3 (30)	6 (60)	9 (45)
Yes, in the past 7 months (*n* (%))	2 (20)	0 (0)	2 (10)

* Data originated from the quantitative survey.

**Table 2 ijerph-19-06098-t002:** Characteristics of participating ED physicians.

	Female *n* = 4	Male *n* = 2	All *n* = 6
Median Age Min–Max	38 27–49	31 29–32	33 27–49
Professional status			
Resident	2	1	3
Specialist	1	1	2
Consultant	1	0	1
Working at 4 EDs in clinics with bed sizes from 350 to 1200			

**Table 3 ijerph-19-06098-t003:** Example of data analysis.

Meaningful Units	Condensed Meanings	Codes
“That really totally works here. I can actually recommend it (ED) everyone. And above all, it’s like this, if you say you have stress with the heart, at the front of the door into this somewhat unpleasant loudspeaker system that trumpets this through the whole room, within three minutes you’re sitting on a bed somewhere and being treated. Someone who just has a knife in his arm, has to wait.”	“I can actually recommend it (ED) everyone. If you have stress with the heart, within three minutes you are treated.”	**Quality of ED treatment**

**Table 4 ijerph-19-06098-t004:** Matrix with main characteristics referring to typical patient patterns.

	Characteristics	A Cardiac Disease (CD) *^,^**	B Mental Health Condition (MHC) *^,^**	C Perceived Quality and Frequency of ED Usage *	D Frequency and Appreciation of Outpatient Care (GP, Specialist) **
Variation					
1		Diagnosed	Diagnosed	Frequent and appreciated	Frequent (regularly and unscheduled)
2		Not diagnosed	Not diagnosed	Pragmatic, reluctant	Pragmatic, scheduled or none

* Data derived from the quantitative survey/medical record. ** Data derived from the qualitative interview data.

**Table 5 ijerph-19-06098-t005:** Presentation of “typical cases”.

Type/*n*	Matrix	Characterization	Quotes
**Type 1** Frequent ED user with CD and MHC *n* = 7	A1 B1 C1 D1	These patients suffer from chronic heart diseases and are burdened by further illnesses, e.g., chronic back pain, renal diseases and/or cancer. They are anxious and resigned. The search for medical treatment plays an important role in their lives on a regular or unregular scheme. The ED is their favored place of rescue when they are overwhelmed by emerging complaints and anxiety.	*“My husband, mostly my husband is not there in the evening, not at home, I am always alone. And when my blood pressure goes up, I call the ambulance” (P17, female, 70 years).* *“When the panic is over and I’ll be lying somewhere, then that’s good, I mean, (…) then you are directly on the site, then you will be helped, yes” (P12, male, 67 years).*
**Type 2** Frequent ED user without CD but with MHC *n* = 2	A2 B1 C1 D1	Patients suffer from recurring fears of a myocardial infarction, even though their heart function is frequently examined. Their PCP visits are often disappointing because they do not cater to their subjective needs. In upcoming perceived emergencies, they head to an ED where they feel taken seriously when presenting with chest pain. They highly appreciate the low-threshold 24/7 availability and the comprehensive medical equipment in EDs.	*“So if I have something in my body that is a bit strange for me, then it is an automatic mechanism that I go to the ED. (…) because there I can be helped most likely and the fastest. The conversation then always calms me down in the ED, much more than at the GP. (…) For me, for example, it is also an emergency if I am only looking for a talk in the ED (…). It helps me get over it” (P04, male, 50 years).*
**Type 3** Needs-based ED user with CD *n* = 7	A1 B2 C2 D2	These patients are characterized by a chronic heart disease which repeatedly requires medical intervention, e.g., cardioversion. They have developed several strategies to cope with their disease and are in regular outpatient treatment. They do not feel very comfortable at the ED but report a pragmatic approach in case of medical needs.	*“I don’t like going to the hospital, although I am actually used to it, on average once or twice a year. And yet I try to avoid that. And that’s why I wait a few days and hope that it will go away on its own. (…) That is now more wishful thinking. But I know that I don’t have to go to the hospital immediately. It’s not life-threatening if you have a pulse of 150 for a few days.” (P01, female, 60 years).*
**Type 4** ED user with CD as alternative to specialist care *n* = 2	A1 B2 C1 D2	This type is an either patient- or PCP-driven regular ED user for acute but not emergency treatment. One patient underlined the fast access and high treatment quality, while another patient who was frequently sent from the GP to the ED complained about the lack of PCP commitments.	*“Specialists? Nah, I don’t have a single one. Because if I feel something, I go immediately to the ED and that’s it. They’re really good doctors here, they’re competent, they think of something and talk to you, wonderful. I can actually recommend it to everybody. And above all, when you say over this somewhat unpleasant loudspeaker in front of the door that you have issues with your heart (…) within three minutes you sit somewhere on a bed being treated” (P14, male, 70 years).*
**Type 5** Surprised by initial cardiac diagnosis in the ED *n* = 2	A1 B2 C2 D1	These patients were newly diagnosed with high blood pressure during routine PCP visits and were urgently referred to the ED by paramedics. ED physicians were the first doctors to explain their unexpected condition and its consequences. Respondents reported their difficulties in coping after discharge and the challenge of making a short-term appointment with a specialist.	*“I was a bit at a loss, because I thought “ok, they’re telling me now what I have to do and what I have to take”, but they said the outpatient doctor has to set it (…). And the letters from the hospital, the instructions were not quite clear” (P05, female, 50 years).* *“You learn through the calls that there are certain keywords that trigger the urgency, when you call in such a practice [cardiologist], call the first one, didn’t trigger, didn’t get an appointment and made a new phrase and the third one has worked, yes” (P10, male, 54 years).*

## Data Availability

Original data are not available for data protection reasons.

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
