# Peer review of "Emergency Departments as Care Providers for Patients with Cardiac Ambulatory Care Sensitive and Mental Health Conditions: Qualitative Interview and Focus Group Study with Patients and Physicians"

_ijerph, 2022, doi:10.3390/ijerph19106098_

Round 1

Reviewer 1 Report

This is nice piece of paper. The primary objective of this qualitative study was to investigate the perception regarding ED use from patients' perspective. The second objective was to triangulate patients' data with the professional view of ED physicians to identify measures to improve the quality of ED care for the specific patient group with somatic illness, such as cardiac diseases.

Mental health condition is an important issue in this study. However, I wonder about the definition of the concept of mental health condition. I cannot relate this concept only to a bad mental health, as everybody have a mental health condition, good or bad. So, I would like a definition of the concept so it would be clear what the authors mean with it.

I really miss a section about previous research, or the body of knowledge, that you want to contribute to. It would be needed to know what we already know, based on previous research, and what you specifically will contribute to.

The conclusion presented in the abstract do not contribute to much new knowledge. All organizations can claim that they need more resources, and here you also claim that the ED also need more resources. It can be interpreted as the authors have fallen into a trap here. It may be the case the the resources could be redistributed, or that more resources not at all improve the situation in the ED healthcare. It should at least be more described how they in that case can provide sustained and successful treatments, and how this should be changed from the current situation.

The introduction miss to describe the aim of the paper. The aim is thus finally presented in the discussion section. Instead the introduction present an objective with a bunch of research questions. I do not think that it is meaningful to have such many research questions in one and the same research article, as it will be very difficult to answer them all in one article. Too many research questions make the article less readable. I think that the authors could try to match the aim or the research question with what really is the conclusions of the article.

A new objective is presented in the end om the materials and methods section, and this is very confusing. It is enough to present the aim of the article in the end of the introduction, after the problematization of the area of interest in the article. If you would combine data in the study, it should not be any objective on its own, instead it would be a method that should be motivated in order to answering the aim of the article.

I would like the authors to describe in more detail what is meant with assigning verbatim parts of the interview both inductively to emergent and deductively to predefined categories. May you can also use a table to give examples.

In line 158 you refer to answering the research questions, but I assume that the methods used will be aimed to answering the research questions, and not just what in claimed here.

I do not fully understand the first sentence in section 3.3. Is it possible to make it more clear?

The results include nice descriptions and quotes.

In the conclusions section I also question the claim that more resources are needed. This is not clear through the article. Everybody also would like to have more resources, always. Moreover, if patients go to the ED instead of for example the primary care has also to do with what they are recommended when they call the healthcare in such emergent situations, see for example: Forsman, B. & Svensson, A. (2019). Frail Older Persons’ Experiences of Information and Participation in Hospital Care, International Journal of Environmental Research on Public Health, Vol. 16, No. 16, Article No. 2829.

Author Response

Response to Reviewer 1 Comments

This is nice piece of paper. The primary objective of this qualitative study was to investigate the perception regarding ED use from patients' perspective. The second objective was to triangulate patients' data with the professional view of ED physicians to identify measures to improve the quality of ED care for the specific patient group with somatic illness, such as cardiac diseases.

Thank you for the in-depth study of our manuscript. In the following, we respond to your comments and questions as best we can.

Point 1: Mental health condition is an important issue in this study. However, I wonder about the definition of the concept of mental health condition. I cannot relate this concept only to a bad mental health, as everybody have a mental health condition, good or bad. So, I would like a definition of the concept so it would be clear what the authors mean with it.

Response 1: We thank the reviewer for bringing this important issue to our attention, which we hereby clarify: According to the American Psychiatric Association a mental health condition (MHC) refers to an abnormal state of mental health with or without a formal diagnosis of mental illness, also called mental disorder, involving significant changes in emotion/mood, thinking and/or behavior and is associated with distress and/or problems functioning in daily social, work or family activities. The term “mental health condition” was introduced in recent years by leading mental health associations (e.g., National Institute of Mental Health, MentalHealth.gov, National Alliance of Mental Illness) to improve language and communication about mental health and to avoid negative labeling, stigma and discrimination. Also, the term emphasizes that a person’s condition can change over time, so a mental health diagnosis might not apply anymore.

To make the term “mental healh conditions” more comprehensible, we added a short explanation and a reference to the manuscript. It now reads: Mental health condition (MHC), which refers to an abnormal state of mental health …. (page 1). Ref: American Psychiatric Association (2021). Words Matter: Reporting on Mental Health Conditions. https://www.psychiatry.org/newsroom/reporting-on-mental-health-conditions).

Point 2: I really miss a section about previous research, or the body of knowledge, that you want to contribute to. It would be needed to know what we already know, based on previous research, and what you specifically will contribute to.

Response 2: We thank the reviewer for the comment. We added information about the previous research and what new findings we will contribute to with our study. It now reads: “Previous research has shown a relation between MHC and CD [24]. Acute anxiety conditions could also be experienced in patients with acute coronary syndrome [25], while depression is connected to poor medical adherence in patients with CDs [26]. These comorbidities pose diagnostic challenges to ED examinations and sustainable treatment. Previous studies focused mainly on younger patients with MHC and cardiac complaints [27-30]. However, more information is needed on older ED patients with cardiac complaint as the probability for CD increases with age [31]. In consequence, MHC comorbidities may remain undiagnosed. To assess the proportion of ED patients who suffer from acute cardiac symptoms and symptoms of MHC, the quantitative part of EMASPOT examined the associations between MHC and increased ED use as well as between MHC and the occurrence of ACSC. The screening of a representative ED patients cohort with cardiac complaints identified a total of 28,4 % with current MHCs [32]. Against the background of these results, complementary qualitative data could reveal a deeper insight into patients’ perspectives and therefore elucidate the role of the ED within their course of treatment [14,33,34]” (page 2).

Point 3: The conclusion presented in the abstract do not contribute to much new knowledge. All organizations can claim that they need more resources, and here you also claim that the ED also need more resources. It can be interpreted as the authors have fallen into a trap here. It may be the case the the resources could be redistributed, or that more resources not at all improve the situation in the ED healthcare. It should at least be more described how they in that case can provide sustained and successful treatments, and how this should be changed from the current situation.

Response 3: We thank the reviewer for the comment. We agree that a general demand for more resources is a poorly targeted conclusion. Therefore, in the revised text, we highlight the need of specific examination measures that can improve care. It now reads: “Our study shows that patients presenting to the ED with cardiac complaints are very different types, many of whom suffer from (co)morbid MHC. Patients with cardiac complaints are treated with high urgency and are examined immediately. That is, for patients with diagnosed and undiagnosed MHC, frequent treatment in the ED can exacerbate MHC when they enter a vicious cycle of complaints-exclusions-complaints. As one consequence, ED personnel should be provided with a tool to detect MHC in patients and explain the bidirectional relationship between physical and mental health problems. Furthermore, improved health literacy by outpatient and inpatient providers could contribute to better patient outcomes and therefore reduce ED care” (page 13).

Point 4: The introduction miss to describe the aim of the paper. The aim is thus finally presented in the discussion section. Instead the introduction present an objective with a bunch of research questions. I do not think that it is meaningful to have such many research questions in one and the same research article, as it will be very difficult to answer them all in one article. Too many research questions make the article less readable. I think that the authors could try to match the aim or the research question with what really is the conclusions of the article.

Response 4: We thank the reviewer for this comment. We now clearly state the research aim in the introduction section and present detailed findings in the results section. Furthermore, we reduced the research questions to the key aspects. It now reads: “The aim of the study on hand is to gain a deeper understanding of the subjective context in which patients with cardiac ACSC seek help in the ED and to analyze possible differences in their approaches. In addition, patient demands should be explored from a professional perspective of ED physicians to develop ideas for improved, targeted care. To achieve the study objective, we defined the following research questions: 1.What different types of patients can be derived with regard to their subjective motivation, personal background and perception of the role of the ED? 2. Which measures would improve the delivery of care from ED physicians’ perspective based on their everyday experiences with patients presenting with cardiac ACSC and assumed MHC?” (page 2).

Point 5: A new objective is presented in the end of the materials and methods section, and this is very confusing. It is enough to present the aim of the article in the end of the introduction, after the problematization of the area of interest in the article. If you would combine data in the study, it should not be any objective on its own, instead it would be a method that should be motivated in order to answering the aim of the article.

Response 5: We thank the reviewer for this helpful comment. We have revised the manuscript and now present the study aim in the introduction section as described in response 4.

Point 6: I would like the authors to describe in more detail what is meant with assigning verbatim parts of the interview both inductively to emergent and deductively to predefined categories. May you can also use a table to give examples.

Response 6: We thank the reviewer for the comment. We now describe in more detail the procedure of data interpretation. It now reads: “A qualitative content analysis (QCA) approach was chosen to reach the study goal and to answer the research questions [43]. QCA is a suitable and transparent method for descriptive qualitative data (see Table 1 for an example of data analysis). The results were worked out by assigning verbatim units of the interviews both inductively to emergent and deductively to predefined categories. In the first stage, one of the authors (MS) reviewed the transcripts and coded them line by line. Then, after several discussions within the research group with professional backgrounds in sociology, psychology and public health (MS, AF, SLK, SO) a coding framework was created. Finally, the main dimensions of the coding structure were condensed to relevant aspects. A “typical case” approach was used to analyze the patients’ interviews [44,45]. Derived from the coding structure, the main narratives about the importance of the ED for health care were identified by condensing meaningful text units into common and distinct patterns derived from the interview data. On this data basis, a matrix was created with the respective variation of cardiac diseases and mental health conditions and the frequency, appreciation and perceived quality of ED and outpatient care (Table 3). By analyzing the content relations and grouping similarities and differences into common and distinct patterns, five different “typical cases” were built. By means of the QCA, the data of the physician focus group were analyzed concerning the improved treatment of patients with MHC. All cases and further results are described in detail and highlighted with meaningful quotes” (page 4).

Furthermore, we followed your suggestion and implemented a table with an example:

Table 1 Example of data analysis.

Meaningful units

Condensed meanings

Code

“That really totally works here. I can actually recommend it (ED) to everyone. And above all, it's like this, if you say you have stress with the heart, at the front of the door into this somewhat unpleasant loudspeaker system that trumpets this through the whole room, within three minutes you're sitting on a bed somewhere and being treated. Someone who just has a knife in his arm, has to wait”.

“I can actually recommend it (ED) to everyone. If you have stress with the heart, within in three minutes you are treated”.

Quality ED treatment

Point 7: In line 158 you refer to answering the research questions, but I assume that the methods used will be aimed to answering the research questions, and not just what in claimed here.

Response 7: We thank the reviewer for this comment. We revised that sentence. It now reads: “By means of the QCA, the data of the physician focus group were analyzed concerning the improved treatment of patients with MHC”(page 4).

Point 8: I do not fully understand the first sentence in section 3.3. Is it possible to make it more clear?

Response 8: We thank the reviewer for this comment. We clarified the sentence. It now reads: “Data show that ED care is more important when satisfaction with outpatient care is low and heart disease is not well managed” (page 8).

Point 9: The results include nice descriptions and quotes.

Response 9: Thank you!

Point 10: In the conclusions section I also question the claim that more resources are needed. This is not clear through the article. Everybody also would like to have more resources, always. Moreover, if patients go to the ED instead of for example the primary care has also to do with what they are recommended when they call the healthcare in such emergent situations, see for example: Forsman, B. & Svensson, A. (2019). Frail Older Persons’ Experiences of Information and Participation in Hospital Care, International Journal of Environmental Research on Public Health, Vol. 16, No. 16, Article No. 2829.

Response 10: We thank the reviewer for referring us to this study, the contents of which we have included in the Discussion and Conclusions sections. It now reads in the Discussion section: “The increased presentation of older patients with complex and chronic conditions and ACSC is one driver of ED crowding, as they require a comprehensive and time-consuming examination. Recent research found out how older patients lack comprehensible information about their health state after discharge and are recommended to ED care inspite of outpatient options [53].” And in the Conclusion section it reads: “. As one consequence, ED personnel should be provided with a tool to identify MHC in patients and explain the bidirectional relationship between physical and mental health problems. Furthermore, improved health literacy by outpatient and inpatient providers could contribute to better patient outcomes and therefore reduce ED care” (page 12).

Reviewer 2 Report

Thank you for submitting the manuscript to this journal. You have chosen an interesting subject to do this study. Some issues need to be clarified. I have summarized these concerns below: the following comments need to be taken with care in order to improve the quality of the manuscript for publishing.

ABSTRACT

The elaboration of the abstract does not adapt to the recommendations of the journal, the subsections (Objectives, methods...) must be eliminated. The abstract contains abbreviations these must be eliminated, the abbreviations must appear in the introduction section, indicating the full name and then the abbreviation. On the other hand, the objective needs to be clarified. the authors must indicate a single general objective of the investigation. Indicate the qualitative design used in the abstract.

KEYWORDS

keywords must be in alphabetical order.

INTRODUCTION

It must appear the aim of the study at the end of the introduction, just before the method.

The introduction is too short, it should be extended. It is important that the authors delve into the current state of knowledge and specify the importance of conducting this study in relation to the existing literature.

The authors should review the introduction to include relevant and current information from the last 5 years. for example check lines 35-37.

There are too old references (2007, 2008, 2009...), it is important that the references are as current as possible. It would be necessary for the authors to carry out a more in-depth analysis of the literature.

Line 71: Review the wording, do not differentiate between study designs in the introduction, what is important is the information provided by these studies.

The objective of the study must be clear and concise, and a single objective.

Questions that the authors pose that are interview questions? Qualitative studies do not have hypotheses. Please review this.

METHODS

This section is incomplete the following subsections are missing, the method must be structured in: design, participants, data collection procedure…

There are aspects that are not clear, such as interviewed participants, how the interviews were carried out, where they were carried out, how the data analysis was carried out.

The design of the study is not clear, what kind of qualitative has been carried out. This is important to define the subsequent analysis of the data.

Some guideline has been followed to determine the quality of the study. In qualitative research it is very important to be rigorous.

Line 114, this information does not apply to this section.

How to guarantee the anonymity of the participants during the analysis of the interviews.

RESULTS

It would be interesting if the authors included a paragraph at the beginning of the results indicating the main themes and sub-themes obtained from the data analysis.

Line 185-187 would be more appropriate to include in the methods section.

The authors could eliminate table 5 and include the citations in each topic.

It would be interesting if the authors included citations in the topics obtained from the focus groups, to justify this interpretation of the data.

It would be interesting to design a figure in which the vision of patients and professionals interrelate.

DISCUSSION

It would be necessary to include new future lines to investigate, after the completion of the study.

CONCLUSION

The conclusions must be clear, direct and respond to the stated objective, for this reason the authors must review them. They are too long, please shorten the conclusions and that they are based on the proposed review.

REFERENCES

The references must be unified and adapted to the standards of the journal.

Author Response

Response to Reviewer 2 Comments

Thank you for submitting the manuscript to this journal. You have chosen an interesting subject to do this study. Some issues need to be clarified. I have summarized these concerns below: the following comments need to be taken with care in order to improve the quality of the manuscript for publishing.

Thank you for reading the manuscript carefully and for your helpful comments, which we took into consideration. Please find our responses below.

Point 1: ABSTRACT The elaboration of the abstract does not adapt to the recommendations of the journal, the subsections (Objectives, methods...) must be eliminated. The abstract contains abbreviations these must be eliminated, the abbreviations must appear in the introduction section, indicating the full name and then the abbreviation. On the other hand, the objective needs to be clarified. the authors must indicate a single general objective of the investigation. Indicate the qualitative design used in the abstract.

Response 1: We thank the reviewer for this comment. We eliminated the subjections and the abbrevations, which now appear first in the Introduction section. Furthermore, we clarified the general objective of the study and we described the evaluation method. It now reads: “The aim of the study on hand is to gain a deeper understanding of the subjective context in which patients with cardiac ACSC seek help in the ED and to analyze possible differences in their approaches. In addition, patient demands should be explored from a professional perspective of ED physicians to develop ideas for improved, targeted care. To achieve the study objective, we defined the following research questions: 1. What different types of patients can be derived with regard to their subjective motivation, personal background and perception of the role of the ED? 2. Which measures would improve the delivery of care from ED physicians’ perspective based on their everyday experiences with patientes presenting with cardiac ACSC and assumed MHC?” (page 2).

Point 2: KEYWORDS keywords must be in alphabetical order.

Response 2: Thank you. The keywords now appear in alphabetical order.

Point 3: INTRODUCTION It must appear the aim of the study at the end of the introduction, just before the method. The introduction is too short, it should be extended. It is important that the authors delve into the current state of knowledge and specify the importance of conducting this study in relation to the existing literature.

Response 3: We thank the reviewer for this comment. We revised the Introduction section comprehensively and added a current state of knowledge. It now reads: “Previous research has shown a relation between MHC and CD [24]. Acute anxiety conditions could also be experienced in patients with acute coronary syndromes such as palpation and chest pain [25]. while depression is connected to poor medical adherence in patients with CDs [26]. These comorbidities pose diagnostic challenges to ED examinations and sustainable treatment. Previous studies focused mainly on younger patients with MHC and cardiac complaints [27-30]. However, more information is needed on older ED patients with cardiac complaint as the probability for CD increases with age[31]. In consequence, MHC comorbidities may remain undiagnosed. To assess the proportion of ED patients who suffer from acute cardiac symptoms and symptoms of MHC, the quantitative part of EMASPOT examined the associations between MHC and increased ED use as well as between MHC and the occurrence of ACSC. The screening of a representative ED patients cohort with cardiac complaints identified a total of 28,4 % with current MHCs [32]. Against the background of these results, complementary qualitative data could reveal a deeper insight into patients’ perspectives and therefore elucidate the role of the ED within their course of treatment [14,33,34]. Moreover, the patients’ view could provide essential information within the development of patient-centered, improved health care services [33]. In addition, patient demands should be explored from a professional perspective of ED physicians to develop ideas for improved, targeted care” (page 2).

Point 4: The authors should review the introduction to include relevant and current information from the last 5 years. for example check lines 35-37. There are too old references (2007, 2008, 2009...), it is important that the references are as current as possible. It would be necessary for the authors to carry out a more in-depth analysis of the literature.

Response 4: We thank the reviewer for this comment. We have revised the Introduction section as described in Response 3 and we included current research published in 2015 and later.

Point 5: Line 71: Review the wording, do not differentiate between study designs in the introduction, what is important is the information provided by these studies.

Response 5: We thank the reviewer for this comment. We have thoroughly revised the Introduction and in this context we referred to the results of the quantitative part of the mixed methods study, which now reads: “To assess the proportion of ED patients who suffer from acute cardiac symptoms and symptoms of MHC, the quantitative part of EMASPOT examined the associations between MHC and increased ED use as well as between MHC and the occurrence of ACSC. The screening of a representative ED patients cohort with cardiac complaints identified a total of 28,4 % with current MHCs [32]. Against the background of these results, complementary qualitative data could reveal a deeper insight into patients’ perspectives and therefore elucidate the role of the ED within their course of treatment [14,33,34]” (page 2).

Point 6: The objective of the study must be clear and concise, and a single objective.

Response 6: We thank the reviewer for this comment. We revised the introduction substantially and focuses the study aim on the overall objective, which now reads: “Objective. The aim of the study on hand is to gain a deeper understanding of the subjective context in which patients with cardiac ACSC seek help in the ED and to analyze possible differences in their approaches. In addition, patient demands should be explored from a professional perspective of ED physicians to develop ideas for improved, targeted care” (page 2).

Point 7: Questions that the authors pose that are interview questions? Qualitative studies do not have hypotheses. Please review this.

Respond 7: We thank the reviewer for this comment. We used semi-structured interview guides, which can be found in the appendix (Table A1 and Table A2). During the interviews respective the focus group, the order of the guidelines content was flexible adapted to the narratives of participants and thematic fields.

Point 8: METHODS. This section is incomplete the following subsections are missing, the method must be structured in: design, participants, data collection procedure…

Respond 8: We thank the reviewer for this comment. We restructered the Methods section as suggested.

Point 9: There are aspects that are not clear, such as interviewed participants, how the interviews were carried out, where they were carried out, how the data analysis was carried out.

Respond 9: We thank the reviewer for this comment. To clarify the sampling, the description of interview conduction and data analysis was revised substantially. It now reads: “2.2. Data collection: patients sampling. Between December 2017 and July 2018, one author (MS) who is experienced in qualitative research in health care settings carried out a purposive sample with n=20 semi-structured face-to-face interviews with patients (interview guide translated into English in Table A1 Appendix). All interviewees had participated in the quantitative EMASPOT survey during a preceding stay in one of the participating eight EDs. A study nurse, who had already explained the study goal in detail, asked about the willingness to take part in a subsequent qualitative interview. The majority of patients agreed. Of these potential participants, 27 were invited by phone to a personal interview. In order to present a broad variation of data, the interviewer approached potential participants with regard to heterogeneity in index clinics, age, gender and main complaints (Table 2) and explained the study goal. Of the requested possible participants declined seven, mostly due to time constraints or because they did not feel well. Place and time of the interview were set by the respondents. According to personal preference, n=11 interviews were carried out at patients’ private homes (n=9) or workplaces (n=2), at the interviewers’ office (n=9), or during a subsequent ED visit (n=1). One interview with a participant who lived in a far distance was conducted by phone. Before starting the data collection, all participants provided written informed consent. The interviewer emphasized that participation was voluntary and could be withdrawn at any time. Interview duration and thematic depth were determined by the respondents. The interviews took from 15 to 52 minutes with a median duration of 30 minutes. Following each interview, field notes were taken to document impressions on atmosphere, nonverbal communication and special features for confirmability. All interview and field notes were transcribed verbatim and entered anonymized into the qualitative data software MAXQDA2020. 2.3 Data collection: physicians sampling After the first evaluation of patients’ interviews, a focus group with a stratified sample of six ED physicians from four study sites was carried out by two researchers with a background in public health and sociology (MS, SO) and one study assistant (interview guide translated into English in Table A2 Appendix) in March 2019. The participants were approached by email or phone with regard to heterogeneity in index clinics, gender, occupational experience and professional position (Table 3). The focus group was conducted as an expert interview and took place in a conference room in one clinic and lasted one hour. [42]. Before starting the discussion, all participants gave written informed consent. 2.4 Data analysis. A qualitative content analysis (QCA) approach was chosen reach the study goal and answer the research questions [43]. QCA is a suitable and transparent method for descriptive qualitative data (example of data analysis in table 1). The results were worked out by assigning verbatim units of the interviews both inductively to emergent and deductively to predefined categories. In the first stage, one of the authors (MS) reviewed the transcripts and coded them line by line. Then, after several discussions within the research group with backgrounds in sociology, psychology and public health (MS, AF, SLK, SO) a coding framework was created. Finally, the main dimensions of the coding structure were condensed to relevant aspects. A “typical case” approach was used to analyze the patients’ interviews [44,45]. Derived from the coding structure, the main narratives about the importance of the ED for health care were identified by condensing meaningful text units into common and distinct patterns derived from the interview data. On this data basis, a matrix was created with the respective variation of cardiac and mental health diseases and the frequency, appreciation and perceived quality of ED and outpatient care (table 3). By analyzing the content relations and grouping similarities and differences into common and distinct patterns, five different “typical cases” were built. With the QCA the data of the physician focus group were analyzed concerning the improved treatment of patients with MHC. All cases and further results are described in detail and highlighted with meaningful quotes” (page 3f).

Point 10: The design of the study is not clear, what kind of qualitative has been carried out. This is important to define the subsequent analysis of the data.

Respond 10: We thank the reviewer for this comment. We revised the manuscript and described in more detail the qualitative content analyses, which we have applied as displayed in response to the point above. Furthermore, we added an small table with an example of the data analysis (page 4).

Point 11: Some guideline has been followed to determine the quality of the study. In qualitative research it is very important to be rigorous.

Response 11: We thank the reviewer for highlighting this important issue. We followed the Consolidated criteria for Reporting Qualitative Research (COREQ) guideslines. Please find attached the completed checklist.

Point 12: Line 114, this information does not apply to this section.

Response 12: We thank the reviewer for this comment. We deleted the information in this section.

Point 13: How to guarantee the anonymity of the participants during the analysis of the interviews.

Respond 13: We thank the reviewer for raising this important issue. We added into the manuscript how we provided data anonymity of our participants. The interviews were transcribed anonymously, that is names and other data that could indicate information about the persons, such as occupations or places where they lived, were not transcribed. Data interpretation was conducted pseudonymized. A list of clear names and pseudonyms is locked and will be deleted ten years after the conduction of the interviews (page 3).

Point 14: RESULTS. It would be interesting if the authors included a paragraph at the beginning of the results indicating the main themes and sub-themes obtained from the data analysis.

Response 14: We thank the reviewer for this comment. We followed the suggestion and included a paragraph at the beginning of the results section. It reads: “Our findings revealed patient types with very different needs against the background of their course of disease(s) and treatment experiences (section 3.1.) Furthermore, patients' data highlight the impact of comorbidities and MHC on the perception of the ED as a site of rescue (section 3.2) which at the same time is intertwined with reported satisfaction with outpatient care and management of the CD (section 3.3). In addition, we present missed organizational support and suggestions for outpatient care improvement from the patient respondents (section 3.4.). The results from the focus group with ED physicians mainly refer to patients with assumed MHC. All discussants were familiar with this type of patient and reported dissatisfaction on the side of patients and professionals because the exclusion of an acute episode does not address an underlying MHC (section 3.5). However, the ED physicians stated the lack of ressources to examine an assumed MHC (section 3.6.) and brought up several suggestions to improve the delivery of care (section 3.7); e.g., an examination tool to assess MHC and refer patients to efficient treatment(section 3.8)” (page 6).

Point 15: Line 185-187 would be more appropriate to include in the methods section.

Respond 15: We thank the reviewer for this comment. We followed the suggestion and included the procedure of the interview conduction into the Methods section (page 8).

Point 16: The authors could eliminate table 5 and include the citations in each topic.

Respond 16: We thank the reviewer for this suggestion, which we followed. The quotes are now included into the Result section.

Point 17: It would be interesting if the authors included citations in the topics obtained from the focus groups, to justify this interpretation of the data.

Response 17: We thank the reviewer for this comment. In the revised manuscript, we highlighted that the results are presented with original citations from the focus group participants, to make the data interpretation transparent and illustrative (page 6).

Point 18: It would be interesting to design a figure in which the vision of patients and professionals interrelate.

Respond 18: We thank the reviewer for this suggestion, which we considered thoroughly. However, we found that the complexity of the results would be lost in a manageable graph. We therefore decided against implementing the suggestion.

Point 19: DISCUSSION It would be necessary to include new future lines to investigate, after the completion of the study.

Respond 19: We thank the reviewer for this comment. We revised the Discussion section and included the need for further research. It now reads: “The results cannot claim generalizability for rural areas. Therefore, further research is needed to obtain results from rural areas and to evaluate the extent to which the application of screening tools for MHC improves patient care” (page 13).

Point 20: CONCLUSION. The conclusions must be clear, direct and respond to the stated objective, for this reason the authors must review them. They are too long, please shorten the conclusions and that they are based on the proposed review.

Respond 20: We thank the reviewer for this comment. We shortened and revised the conclusions to make them more clear. It now reads: “Our study shows that patients presenting to the ED with cardiac complaints are very different types, many of whom suffer from (co)morbid MHC. Patients with cardiac complaints are treated with high urgency and are examined immediately. That is, for patients with diagnosed and undiagnosed MHC, frequent treatment in the ED can exacerbate MHC when they enter a vicious cycle of complaints-exclusions-complaints. As one consequence, ED personnel should be provided with a tool to detect MHC in patients and explain the bidirectional relationship between physical and mental health problems. Furthermore, improved health literacy by outpatient and inpatient providers could contribute to better patient outcomes and therefore reduce ED care” (page 13).

Point 21: REFERENCES. The references must be unified and adapted to the standards of the journal.

Respond 21: We thank the reviewer for this hint. We have revised the references accordingly by using the MDPI style in EndNote.

Round 2

Reviewer 2 Report

Congratulation.